# Melatonin-Induced Postconditioning Suppresses NMDA Receptor through Opening of the Mitochondrial Permeability Transition Pore via Melatonin Receptor in Mouse Neurons

**DOI:** 10.3390/ijms23073822

**Published:** 2022-03-30

**Authors:** Takanori Furuta, Ichiro Nakagawa, Shohei Yokoyama, Yudai Morisaki, Yasuhiko Saito, Hiroyuki Nakase

**Affiliations:** 1Department of Neurosurgery, Nara Medical University, Shijocho 840, Kashihara 634-8521, Japan; f_ta30@yahoo.co.jp (T.F.); shoheidon1182@gmail.com (S.Y.); dueni5714@yahoo.co.jp (Y.M.); nakasehi@naramed-u.ac.jp (H.N.); 2Department of Neurophysiology, Nara Medical University, Shijocho 840, Kashihara 634-8521, Japan; syasu@naramed-u.ac.jp

**Keywords:** ischemic postconditioning, melatonin receptor, NMDA receptor, mitochondrial permeability transition pore

## Abstract

Mitochondrial membrane potential regulation through the mitochondrial permeability transition pore (mPTP) is reportedly involved in the ischemic postconditioning (PostC) phenomenon. Melatonin is an endogenous hormone that regulates circadian rhythms. Its neuroprotective effects via mitochondrial melatonin receptors (MTs) have recently attracted attention. However, details of the neuroprotective mechanisms associated with PostC have not been clarified. Using hippocampal CA1 pyramidal cells from C57BL mice, we studied the involvement of MTs and the mPTP in melatonin-induced PostC mechanisms similar to those of ischemic PostC. We measured changes in spontaneous excitatory postsynaptic currents (sEPSCs), intracellular calcium concentration, mitochondrial membrane potential, and N-methyl-D-aspartate receptor (NMDAR) currents after ischemic challenge, using the whole-cell patch-clamp technique. Melatonin significantly suppressed increases in sEPSCs and intracellular calcium concentrations. The NMDAR currents were significantly suppressed by melatonin and the MT agonist, ramelteon. However, this suppressive effect was abolished by the mPTP inhibitor, cyclosporine A, and the MT antagonist, luzindole. Furthermore, both melatonin and ramelteon potentiated depolarization of mitochondrial membrane potentials, and luzindole suppressed depolarization of mitochondrial membrane potentials. This study suggests that melatonin-induced PostC via MTs suppressed the NMDAR that was induced by partial depolarization of mitochondrial membrane potential by opening the mPTP, reducing excessive release of glutamate and inducing neuroprotection against ischemia-reperfusion injury.

## 1. Introduction

The phenomenon of acquiring ischemic tolerance by intermittent ischemic stress before fatal ischemia is called ischemic preconditioning and has been shown to have marked neuroprotective effects on cerebral ischemia-reperfusion injury [1,2,3]. However, the clinical application of ischemic preconditioning to acute ischemic strokes (AIS) is not realistic unless the onset of an AIS can be predicted. Intermittent ischemic loading after severe ischemia has also been shown to inhibit ischemia-reperfusion injury, which is called ischemic postconditioning (PostC) [4,5]. Since the timing of reperfusion after an AIS is clinically recognizable, the concept of PostC may be applicable as a new therapeutic tool in addition to intravenous plasminogen activator (tPA) treatment and mechanical thrombectomy. One of the key cellular processes involved in ischemia-reperfusion injuries in the brain is a disruption of the N-methyl-D-aspartate receptor (NMDAR). That is, excessive activation of the NMDAR leads to an excessive increase in cytoplasmic Ca^2+^ concentrations and activation of proteins such as caspases and endonucleases [6]. Another important factor involved in the ischemia-reperfusion injury is the mitochondrial permeability transition pore (mPTP). Although the mPTP regulates mitochondrial function, the opening of the mPTP leads to eventual cell death, apoptosis, or necrosis [7]. The ischemia-reperfusion injury induced excessive calcium accumulation, ROS production, and ATP depletion, leading to the opening of the mPTP, which is a key event in cell death caused by ischemia-reperfusion injury [8,9]. Therefore, the mPTP is indispensable in elucidating the mechanism of neuroprotection against ischemia-reperfusion injury. We have previously reported that the opening of mitochondrial ATP-dependent potassium (mito-K_ATP_) channels is involved in the inhibitory effect of ischemic PostC on the synaptic over-release of glutamate, and that ischemic PostC suppresses Ca^2+^ influx into the cytoplasm by reducing NMDAR-mediated currents through the opening of the mPTP. We also found that the depolarization of the mitochondrial membrane potential by opening the mito-K_ATP_ channel is central to the mechanism of ischemic PostC in the neuroprotection against anoxic injury [10,11]. Previous animal studies have shown that diazoxide, a mito-K_ATP_ channel opener, has a neuroprotective effect against cerebral infarction [10,11,12,13,14]; however, there are problems with the clinical use of diazoxide for cerebral infarction due to adverse effect, such as hyperglycemia, heart failure, and edema [15]. Because of these properties, diazoxide is not an ideal drug for the treatment for AIS.

Melatonin is an endogenous hormone that regulates circadian rhythms and has multifaceted effects such as anti-oxidation, anti-inflammation, anti-apoptosis, and reduction of autophagic cell death [16,17,18,19,20], as well as being very safe to administer to the human body [21]. The melatonin receptors (MTs) are located in the brain plasma membranes and in the outer mitochondrial membranes [22]. In fact, the neuroprotective effects of melatonin have already been reported [23]. Melatonin rapidly activates various signaling cascades through the MTs and exerts various effects. Of the three subtypes of MTs (MT1, MT2, and MT3), only MT1 and MT2 are expressed in mammals [16]. In recent years, the neuroprotective effects of melatonin have been shown to be at least partially due to the activation of MT1 and MT2 in the ischemic brain [24]. The MT activation induces a variety of signaling cascades, in turn inducing neuroprotective effects by reducing ischemia-induced inflammation, oxidative stress, and mitochondrial dysfunction [25]. These findings suggest the potential for application as a new therapeutic approach in the treatment of ischemic strokes [26]. However, there are few reports of melatonin-induced PostC in neurons [27], and the mechanism of melatonin-induced PostC in neurons is still unclear. Previous studies have not examined the involvement of the mPTP, Ca^2+^ dynamics, NMDAR behavior, or glutamate changes in the mechanism of melatonin-induced PostC [27].

In order to examine the efficacy and detailed mechanism of melatonin-induced PostC, we analyzed changes in spontaneous excitatory postsynaptic currents (sEPSCs), NMDAR currents, cytosolic Ca^2+^ concentrations, and mitochondrial membrane potentials under melatonin-induced PostC in the hippocampal cornu ammonis (CA1) pyramidal neurons using the whole-cell patch-clamp technique.

## 2. Results

In this study, we examined the effect of melatonin-induced PostC after cerebral ischemia using the same oxygen–glucose deprivation model as used in previous experiments [10,11]. We randomly assigned mouse hippocampal slices to the following groups (Figure 1) and examined the following items.

### 2.1. Melatonin-Induced PostC Suppresses the Surge of sEPSCs

In order to verify whether melatonin-induced PostC can indicate neuroprotective effects in the whole-cell recording experiment, we first examined whether the melatonin group could suppress the surge of sEPSCs caused by ischemia-reperfusion in the control group. The frequency of sEPSCs gradually increased during the second half of the ischemia-perfusion period at 7.5 min of ischemic load, sharply increased after reperfusion, then gradually decreased [10]. To determine the optimal concentration of melatonin in this experimental setting, three different concentrations of melatonin were perfused after 7.5 min of ischemia: low (10 μM; *n* = 5); medium (100 μM; *n* = 5); and high (1 mM; *n* = 5). Melatonin-induced PostC suppressed this rapid increase in sEPSCs frequencies (Figure 2A,B). The maximum effect of melatonin on the suppression of sEPSCs was observed at 100 μM (Figure 2A,B). The percentage of cumulative sEPSCs at 20 min of melatonin-induced PostC group (Mel, *n* = 5) after the onset of ischemic perfusion were significantly lower than that of the control group (Con, *n* = 5) (Mel, 9.29 ± 1.95 × 103%; Con, 25.4 ± 2.82 × 103%, *p* < 0.01) (Figure 2C).

### 2.2. Melatonin-Induced PostC Reduces the Number of Dead Hippocampal CA1 Neurons

According to a previous report [10], the number of dead cells were counted using two different dyes at two different time points—to exclude the effect of cells dying during the slice preparation process and to evaluate only cells dying due to ischemic load (Figure 3A). We found that there were significantly fewer dead cells after ischemia in the melatonin-induced PostC group than in the control group (Mel, *n* = 11, 86 ± 2.65/mm CA1; Con, *n* = 11, 138 ± 5.94/mm CA1, *p* < 0.01) (Figure 3B). This finding indicated that melatonin-induced neuroprotection mediated by PostC acts within 20 min after ischemia-reperfusion injury.

### 2.3. Melatonin-Induced PostC Silences NMDAR Currents after Reperfusion

When NMDA was applied to CA1 pyramidal cells with a puff, an inward current consisting of a fast-falling phase and a slow-decaying phase persisted for several seconds (Figure 4A). In both ischemia and chemical PostC, NMDAR currents decreased in the early stage of perfusion. The change in multiples of Amps units of NMDAR current amplitude between 10 and 20 min after reperfusion was compared between control, melatonin, the MT agonist, ramelteon (Ram), the MT antagonist, luzindole (Luz) + melatonin, and the mPTP inhibitor, cyclosporine A (CsA), + melatonin groups. NMDA-induced currents in the early stage of reperfusion were greatly reduced in the melatonin group (*n* = 10) compared to the control group (*n* = 10) (Con, 1.07 ± 0.0824 vs. Mel, −0.622 ± 0.0532, *p* < 0.01) (Figure 4B). In the ramelteon group (*n* = 10), NMDA-induced currents were reduced compared to the control group (Control, 1.07 ± 0.0824 vs. Ram, 0.653 ± 0.0403, *p* < 0.01) (Figure 4B). Furthermore, in the melatonin group, NMDA-induced currents were decreased compared to the luzindole and melatonin group (*n* = 9) and melatonin and CsA group (*n* = 9) (Mel, 0.622 ± 0.0532 vs. Luz + Mel, 1.11 ± 0.0679, *p* < 0.01; Mel, 0.622 ± 0.0532 vs. CsA + Mel, 1.09 ± 0.127, *p* < 0.01) (Figure 4B). These results indicate that melatonin-induced PostC suppresses NMDAR activity in the early stage of reperfusion and that the effect is mediated by MTs. In addition, CsA abolishes the inhibitory effect of melatonin on NMDAR activity.

### 2.4. Postconditioning Suppresses Cytosolic Ca^2+^ Increase

To evaluate the involvement of cytosolic Ca^2+^ in the neuroprotection afforded by melatonin-induced PostC, we examined changes in cytosolic Ca^2+^ in the control and melatonin-induced PostC groups. During the anoxic period, the Fura-2 ratio gradually increased, indicating an increase in cytosolic Ca^2+^ (Figure 5A,B). Cytosolic Ca^2+^ continued to increase until 5 min after anoxia, then gradually decreased (Figure 5B). The change in the Fura-2 ratio between 5 and 10 min after anoxia was analyzed between groups. In the melatonin-induced PostC group (*n* = 9), the rate of change in the Fura-2 ratio was significantly lower than that in the control group (*n* = 14) (Con, 16.7 ± 3.33% vs. Mel, 6.74 ± 2.04%, *p* < 0.05) (Figure 5C). These results indicate that melatonin-induced PostC inhibits the elevation of cytosolic Ca^2+^ in the early stage of reperfusion.

### 2.5. Mitochondria Temporarily Depolarize during Melatonin-Induced PostC

Changes in mitochondrial membrane potential were examined in the control, melatonin, ramelteon, and luzindole + melatonin groups. The green/red ratio, which represents depolarization of the mitochondrial membrane potential and obtained via JC1 fluorescence, began to increase 5 min after the start of the ischemic load and continued to increase until 3 min after the start of reperfusion (until 2 min after the start of reperfusion in the ramelteon group), then decreased in all four groups (Figure 6A). The rate of change in the green/red ratio between 7.5 and 12.5 min after the start of reperfusion was compared among the four groups. The green/red ratio in the melatonin group (*n* = 6) and ramelteon group (*n* = 5) was significantly higher than that in the control group (*n* = 6) (Con, 0.915 ± 1.3% vs. Mel, 12.2 ± 2.06%, *p* < 0.05; Con, 0.915 ± 1.3% vs. Ram, 6.22 ± 0.554%, *p* < 0.05), and the green/red ratio in the melatonin group (*n* = 6) was significantly higher than in the luzindole + melatonin group (*n* = 5) (Mel, 12.2 ± 2.06% vs. Luz + Mel, 0.711 ± 1.05%, *p* < 0.05) (Figure 6B). These results indicate that the mitochondrial membrane potential was more depolarized in the melatonin group than in the control group in the early stage of reperfusion. These results also suggested that the change in mitochondrial membrane potential after reperfusion was due to MT-mediated recruitment.

## 3. Discussion

The present study showed the neuroprotective effect of melatonin-induced PostC against ischemia-reperfusion injuries in mouse hippocampal CA1 cells, using an electrophysiological approach. Melatonin reduced a surge of synaptic glutamate release and neuronal cell death after ischemia-reperfusion. The melatonin-induced PostC strongly suppressed the increase in the intracellular Ca^2+^ concentration under anoxic conditions by downregulating the NMDARs. Both the mPTP inhibitor (CsA) and the MT blocker (luzindole) abolished melatonin-induced PostC. Furthermore, both melatonin and the melatonin agonist depolarized the mitochondrial inner membrane potential in the early phase of ischemia-reperfusion. These results suggest that MTs play an important role in neuroprotection after ischemia-reperfusion injury mediated by PostC mechanisms.

### 3.1. Melatonin-Induced PostC Reduces Excessive Accumulation of Extracellular Glutamate

During brain ischemia and ischemia-reperfusion, an excessive release of glutamate is observed, triggering neuronal death [28,29,30]. The excess glutamate release results in over-activation of NMDARs and leads to a Ca^2+^ overload inside the neurons. This intracellular Ca^2+^ overload triggers a range of downstream pro-death signaling events such as calpain activation, reactive oxygen species (ROS) generation, and mitochondrial damage, resulting in cell necrosis or apoptosis [31,32,33]. Therefore, inhibiting the excessive release of glutamate after ischemia-reperfusion is important for neuroprotection against ischemia-reperfusion injury. Yokoyama et al. reported that ischemic PostC suppressed the surge of glutamate release during the immediate–early reperfusion period after ischemic insult and described the opening of the mito-K_ATP_ channels as being essential for suppressing the surge in released glutamate [10]. Herrera et al. reported that melatonin prevents glutamate-induced oxytosis in the HT22 mouse hippocampal cell line through an antioxidant effect, specifically targeting the mitochondria [34]. In the present study, melatonin suppressed the surge of glutamate during the early reperfusion period after ischemia and reduced the histological neuronal cell death. These results indicate that the administration of melatonin after ischemia-reperfusion suppressed an excessive release of glutamate and reduced neuronal death following ischemia-reperfusion injury. The clearance of the extracellular glutamate is important for maintaining a low extracellular glutamate concentration and the homeostasis of the central nervous system. This process is largely mediated by two types of glutamate transporters which are mainly expressed in astrocytes, excitatory amino acid transporter (EAAT)-1/glutamate-aspartate transporter (GLAST), and EAAT-2/glutamate transporter-1 (GLT-1) [34,35]. Although glutamate transporters play an important role in regulating the synaptic glutamate concentration in cerebral ischemia, some studies showed that the expression of GLT-1 was decreased in the early period of ischemia [36,37]. Whether the glutamate transporters play a neuroprotective or neurodegenerative role in regulating glutamate levels in the synaptic cleft during cerebral ischemia is still controversial, as it has been suggested that the role of glutamate transporters may vary depending on the timing, extent, and location of cerebral ischemia [38]. It is unclear how melatonin acts on the glutamine transporter during cerebral ischemia, and further research is needed.

### 3.2. Melatonin Suppresses Influx of Extracellular Ca^2+^ by Downregulating NMDARs after Ischemia

Ischemic stress in neurons leads to the accumulation of intracellular Ca^2+^ [31]. This increase in cytosolic calcium is transmitted to the mitochondrial matrix via a Ca^2+^ channel, the mitochondrial Ca^2+^ uniporter (MCU) [39], located on the inner mitochondrial membrane. Excessive increases in the matrix Ca^2+^ alter the permeability of the mitochondrial membranes, impairing their ability to generate ATP and causing a release of pro-apoptotic factors [31,40,41]. Mitochondrial dysfunctions due to calcium overloads are important in the process of ischemia-induced cell death [42,43]. In the present study, we observed that the intracellular Ca^2+^ started to increase after ischemia-reperfusion, not during ischemia, and remained at higher levels than in the pre-ischemia period for 20 min. However, we found that melatonin-induced PostC suppressed these increases in intracellular Ca^2+^ after ischemia-reperfusion. The NMDARs play a dominant role in mediating the glutamate-induced lethal Ca^2+^ influx [44]. We have previously reported that the NMDAR functions as the primal gate for the Ca^2+^ influx during the early reperfusion period [11]. In the present study, we found that the melatonin-induced PostC reduced the amplitude of the whole-cell inward current induced by the NMDAR puff after ischemic stress. These results suggest that melatonin acts on the NMDARs through an intracellular mechanism to reduce NMDAR conductivity, thereby disrupting the positive feedback loop of the Ca^2+^ influx and preventing the increase in the intracellular Ca^2+^ concentration that persists after an ischemic insult. Escames et al. reported that two major actions of melatonin may be directly related to the suppression of the NMDAR-mediated excitation without involving MTs: (i) the reduction of neuronal nitric oxide (NO) synthase activity and thus reduction of NO production and (ii) the regulation of the redox site of the NMDARs [45]. In contrast, the present study proved that the MT antagonist, luzindole, inhibited the effect of melatonin-induced PostC on NMDAR down-regulation, suggesting that NMDAR down-regulation can be mediated via MTs. Furthermore, CsA, an mPTP inhibitor, reduced the NMDAR activity. It is reported that the mPTP is involved in the down-regulation of NMDARs [6]. Taken together, in addition to the direct effect of melatonin on NMDARs, the function of the mPTP via MTs may play an important role in the mechanism of NMDAR down-regulation in melatonin-induced PostC.

### 3.3. Melatonin Leads to Neuroprotection by Putting mPTP into Low-Conductance Mode

The mPTP is associated with both apoptosis and necrosis, and also plays an important role in the regulation of mitochondrial functions by regulating the mitochondrial membrane potential, calcium homeostasis, ROS production, and ATP production [46,47,48,49]. Mitochondrial dysfunction is an underlying cause of ischemia-reperfusion injury. Ischemia-reperfusion injury induces greatly increasing ROS generation and calcium overload, eventually triggering the opening of the mPTP [8,9]. The opening of the mPTP results in the free passage of low molecular mass solutes (<1500 Da) and mitochondrial apoptotic protein, cytochrome c, across the inner mitochondrial membrane [9,50,51]. Under these conditions, the mitochondrial membrane potential is dissipated, leading to ATP hydrolysis by the reversal of the F_0_F_1_-ATP synthase and the consequent cellular energy depletion, eventually resulting in cell death [9]. Recent studies have suggested that the F_0_F_1_-ATP synthase C subunit of mitochondria is a major component of the mPTP [52]. Cyclophilin (Cyp) D blinds the lateral stalk of the F_0_F_1_-ATP synthase and positively regulates the pore opening [53,54]. It should be noted that CypD is a mitochondrial receptor in CsA, and although CsA can desensitize the mPTP via CypD, it does not always inhibit the mPTP from opening [55,56]. However, in experimental animals, the administration of CsA decreases cerebral infarct volume by inhibiting the mPTP from opening [57]. Furthermore, inhibition of the mPTP reportedly provides neuroprotection against cerebral ischemia [58,59]. The exact mechanism of mPTP regulation against ischemia-reperfusion injury has remained unclear. In our experiments, the inhibition of mPTP opening after administration of CsA cancelled the effects of the melatonin-induced PostC, leading to an increase of NMDAR currents after reperfusion. This result appears to differ from other previous findings [57] but is consistent with our previous report [11]. Okahara et al. reported that both the mPTP opening and inflammation are necessary to improve the neurological outcomes after cerebral ischemia-reperfusion injury on experiments using cyclophilin D and CC chemokine receptor 2-knockout mice [60]. Hawrysh et al. reported the mechanism of tolerance against anoxia in turtle neurons, finding that anoxia activates mito-K_ATP_ channels, leading to matrix depolarization and triggering the transient opening of the mPTP and Ca^2+^ release via the mPTP, and ultimately silencing the NMDARs in turtle neurons [6]. Such findings are consistent with the present results. Although a high-conductance-mode opening allows the passage of ions that leads to cell death, the mPTP also exhibits a transient, low-conductance-mode opening that contributes to Ca^2+^ homeostasis and the regulation of mitochondrial function [61]. These results suggested that melatonin may induce the opening of a low-conductive mPTP. As a result, Ca^2+^ can be released from the mitochondria matrix via the mPTP, eventually inducing down-regulation of the NMDAR.

### 3.4. Role of Melatonin Receptors in Melatonin-Induced PostC

Various lines of evidence have been reported on the neuroprotective effects of melatonin against cerebral ischemia [23,26,62,63]. Although MTs are a major target of melatonin, whether melatonin prevents damage caused by cerebral ischemia via the MT remains controversial. The exact mechanisms underlying the neuroprotective effects of melatonin are unknown but might be attributed to its radical scavenging and antioxidant properties [64,65]. Kilic et al. reported that the neuroprotective effect of melatonin is not MT-mediated, as the effects of melatonin on infarct volume and edema reduction in the middle cerebral artery occlusion model did not differ significantly between wild-type and MT1/MT2-knockout mice [66]. However, Shaida et al. suggested that the direct inhibition of the mPTP by melatonin may essentially contribute to its anti-apoptotic effects in transient brain ischemia [67]. In the present study, melatonin reduced the NMDAR-induced current following ischemia-reperfusion insult, and luzindole (when combined with melatonin) abolished the neuroprotective effect. In addition, luzindole significantly decreased the mitochondrial membrane potential after ischemia-reperfusion. Furthermore, the MT agonist, ramelteon, induced mitochondrial membrane potential depolarization after ischemia-reperfusion, similar to melatonin. Since ramelteon itself does not have the radical scavenging and antioxidant effect that melatonin has [68], our results suggest that melatonin-induced PostC mainly acts on the mPTP via MTs. A recent report proved that melatonin-mediated neuroprotection results from the binding of melatonin to the mitochondrial MT [22]. Wu et al. reported that the neuroprotective effect of ramelteon may be attributed to its agonism on MTs, and its inhibition of autophagy in ischemic brains in mice via the MCAO model [27]. These reports suggest that MTs are required for melatonin-induced neuroprotection against ischemia-reperfusion and support our findings. On the other hand, there was a slight difference in the effect of mitochondrial depolarization between melatonin and ramelteon in this experiment. This may be related to the non-MT-mediated effects of melatonin. Further studies are needed to elucidate the exact mechanism of how melatonin acts on the mPTP to induce the neuroprotective effect against ischemia-reperfusion injury.

### 3.5. Conductance Control of mPTP and Melatonin-Induced PostC Mechanism

The methods for regulating the mPTP opening in low-conductance mode and high-conductance mode are mainly the mitochondrial Ca^2+^ and the inner mitochondrial membrane potential (ΔΦ) [61]. The long-lasting opening of the mPTP in high-conductance mode allows for the passage of ions, including Ca^2+^, leading to the dissipation of ΔΦ and eventually resulting in cell death. On the other hand, the mPTP can exhibit opening in low-conductance mode that contributes to Ca^2+^ homeostasis and the regulation of mitochondrial function [48]. The switch between modes is imposed by ΔΦ, with the threshold value being controlled by mitochondrial Ca^2+^ [69]. In the present study, melatonin depolarized the ΔΦ after ischemia-reperfusion injury. This suggests that anoxia stops the proton pump and inhibits rapid recovery from the depolarized state to the matrix-negative membrane potential. This result is similar to our previous findings on ischemic PostC [11]. We hypothesize that melatonin acts on the mPTP via MTs and protects the mPTP from high-conductance mode. The MCU may play an important role in this function. Although the ischemia-reperfusion injury results in high levels of cytosolic Ca^2+^, mitochondrial Ca^2+^ is assumed to be equivalent to the cytosolic concentration because the driving force for Ca^2+^ by ΔΦ is lost. When the pO2 and glucose concentrations increase to normal levels during reperfusion, mitochondrial respiration begins, polarizing the inner membrane and generating ATP. As the negative potential of the matrix is restored, Ca^2+^ is taken up through the MCU. At this point, the cell membrane is depolarized, and glutamate is accumulated extracellularly due to the ischemic reperfusion injury. The Ca^2+^ is recruited into the cell via NMDAR and continues to be transferred to the mitochondrial matrix by the driving force of the ΔΦ. Finally, the mPTP is thought to open to release excess Ca^2+^ in the mitochondrial matrix. The melatonin-induced PostC leads to the depolarization of the ΔΦ, reducing the driving force for Ca^2+^ by the ΔΦ to the mitochondrial matrix and thereby avoiding excessive accumulation of Ca^2+^ in the matrix. This in turn prevents the mPTP from opening in high-conductance mode, resulting in the down-regulation of NMDAR and the suppression of the glutamate surge. This may lead to neuroprotection.

The present study was limited to the very early phase after ischemia-reperfusion, observing only the first 20 min after ischemia-reperfusion. Before melatonin can be applied clinically to the acute phase of cerebral infarction, the timing of administration and the longer-term course after cerebral infarction need to be confirmed using in vivo experiments. In the present study, we indicated that MTs are involved in melatonin-induced PostC, but we may not be able to show enough direct evidence to elucidate how the MTs are involved in the neuroprotective mechanism. One possibility is that the MTs lead to the opening of the mito-K_ATP_ channel, which has been reported in rat hearts [68]. However, it is still unclear how the MTs open the mito-KATP channel, and further studies are needed.

## 4. Materials and Methods

### 4.1. Preparation of Mouse Hippocampal Slices

All experimental procedures were approved by the Animal Care and Use Committee of the Nara Medical University (approval no. 12599) and were performed in accordance with the Guidelines for the Proper Conduct of Animal Experiments. Wild mice 4- to 8-weeks-old C57BL/6J (65 males), weighing 18–24 g were used for the experiments. Mice were maintained on a 12:12 h light cycle and had free access to food and water. Mice were anesthetized with isoflurane (0.05 *v/v*, given by inhalation), then killed by decapitation. The brains were quickly removed and immersed in ice-cold solution (composition: sucrose 230 mM, KCl 2.5 mM, NaHCO_3_ 25 mM, NaH_2_PO_4_ 1.25 mM, CaCl_2_ 0.5 mM, MgSO_4_ 10 mM, and D-glucose 10 mM) then heated with 95% O_2_/5% CO_2_. By means of a linear slicer (PRO7; DOSAKA EM, Kyoto, Japan), horizontal slices of the hippocampal formation and adjacent cortex were cut in the above solution at a thickness of 350 μm. Slices were then incubated in standard artificial cerebrospinal fluid (aCSF) (composition: NaCl 125 mM, KCl 2.5 mM, NaHCO_3_ 25 mM, NaH_2_PO_4_ 1.25 mM, CaCl_2_ 2.0 mM, MgCl_2_ 1.0 mM, and D-glucose 10 mM) and bubbled with the same gas mixture for at least 1 h at 32 °C. Cells were then kept in aCSF at 27 °C.

### 4.2. Patch-Clamp Recording

Individual slices were placed in an 800 µL recording chamber that was continually perfused with gas-saturated aCSF at a flow rate of 2.0 mL/min. The temperature was kept between 31–33 °C by a controlled heater attached to the inlet. The recording chamber was mounted on a BX50WI vertical microscope (Olympus, Tokyo, Japan) fitted with an infrared differential interference microscope and epifluorescence imaging equipment. Recordings of whole-cell voltage-clamps were performed from visually confirmed CA1 pyramidal cell somas using an EPC-9 patch-clamp amplifier (Heka, Lambrecht/Pfalz, Germany). The holding potential was set at −70 mV. Patch pipettes were made of thick-walled borosilicate glass capillaries and filled with an internal solution containing Cs-gluconate (141 mM), CsCl (4.0 mM), MgCl_2_ (2.0 mM), HEPES (10.0 mM), Mg-ATP (2.0 mM), Na-GTP (0.3 mM), and EGTA (0.2 mM) (pH 7.25), and filled with an internal solution containing CsOH. For analyzing EPSCs, a solution containing K-gluconate (141 mM), KCl (4.0 mM), MgCl_2_ (2.0 mM), HEPES (10.0 mM), Mg-ATP (2.0 mM), Na-GTP (0.3 mM), EGTA (0.2 mM) (pH 7.25) and KOH was used. The resistance of the pipette was 2.5–3.5 MΩ. If the access resistance exceeded 20 MΩ, whole-cell recordings were rejected. To isolate glutamatergic EPSCs, all recordings were made in aCSF supplemented with the GABA_A_ and GABA_B_ antagonist picrotoxin (50 µM).

### 4.3. Simulating Ischemia and Pharmacological Postconditioning in Hippocampal Slices

Severe cerebral ischemia was simulated by exposing hippocampal slices to a solution in which glucose and oxygen were replaced by sucrose and nitrogen. After 7.5 min of ischemia, reperfusion was performed for 20 min [10,11]. Pharmacological PostC was initiated after 7.5 min of ischemia, and hippocampal slices were reperfused with melatonin for 20 min [10,11]. Melatonin, ramelteon, luzindole, CsA, and picrotoxin were purchased from Sigma-Aldrich (St. Louis, MO, USA).

### 4.4. Perfusion Protocols

Slices of mouse hippocampus were randomly assigned to one of the 5 following groups. Each group underwent a baseline period of 5 min of normoxia, 7.5 min of ischemia, and 20 min of reperfusion, respectively. After the ischemic insult and with the onset of reperfusion, the different experimental groups underwent the following protocols (Figure 1): (1) control group (Con), slices were perfused with aCSF for 20 min; (2) melatonin group (Mel), slices were perfused with aCSF containing 100 µM of melatonin for 20 min; (3) ramelteon group (Ram), slices were perfused with aCSF containing 100 µM of the MT agonist, ramelteon, for 20 min; (4) luzindole and melatonin group (Luz + Mel), slices were perfused with aCSF containing melatonin in combination with the MT antagonist, luzindole (100 µM), for 20 min; (5) CsA and melatonin group (CsA + Mel), slices were perfused with aCSF containing melatonin in combination with the mPTP inhibitor, CsA (2 µM), for 20 min. The doses of ramelteon and luzindole were aligned with those of melatonin [68]. The dose of CsA was the same as in our previous study [11].

### 4.5. Recording of Whole-Cell Current Responses to NMDA Application

To assess the sensitivity of NMDARs, whole-cell current responses to NMDA application were recorded. NMDA (5 µM) was puffed to the cell body for 80–160 ms with a micropipette similar to that used for whole-cell recordings. Low gas pressure was applied (nitrogen, 4–6 psi) to the puff micropipette, and gas pressure was kept constant throughout the recording. To suppress Mg^2+^ blocking of NMDAR channels, the neuron was voltage-clamped to a holding potential of −55 mV during the pre- (1 s) and post-stimulation period (6 s). Experiments were performed over 32.5 min (up to 20 min after reperfusion), with NMDAR currents recorded every 30 s.

### 4.6. Fluorometric Evaluation of Cytosolic Ca^2+^ Changes

To assess Ca^2+^ changes in cytoplasm, 15 µmol/L of Fura-2 (DOJINDO, Kumamoto, Japan) was added to the pipette solution and the Fura-2 fluorescence signal of whole-cell voltage-clamped pyramidal neurons was measured. Fura-2 was excited every 10 s at 340 nm and 380 nm using a fast-switching multi-wavelength illumination system (Lambda DG-4; Sutter Instruments, Novato, CA, USA). Fluorescence emission was long-pass filtered at 510 nm, and a 500 nm dichroic mirror was used. A × 40 water immersion objective lens (LUMPlanFI/IR, Olympus, Tokyo, Japan) and a CCD camera (CoolSNAP EZ; Photometrics, Tucson, AZ, USA) were used to acquire images. Illumination and image acquisition were regulated by MetaMorph software (Molecular Devices, San Jose, CA, USA). The region of interest (ROI) was defined as a circular area (5 µm in diameter) with maximum fluorescence intensity located near the center of the somatic cell. The ratio of mean fluorescence intensity (340 nm excitation/380 nm excitation) in the ROI was computed.

### 4.7. Fluorometric Evaluation of Mitochondrial Membrane Potential

To evaluate mitochondrial membrane potential, JC1 (Cayman Chemical, Ann Arbor, MI, USA), a fluorescent dye for which the emission wavelength changes depending on the membrane potential, was loaded into the cytoplasm via a patch pipette. The patch pipette was filled at the tip with an internal solution containing no dye and returned with an internal solution containing dye (2.0 µM) quickly before use. The J-aggregated state (red fluorescence) of JC1 was excited at 548 nm using a 580 nm dichroic mirror and fluoresced at 590 nm with a long-pass filter. The monomeric state (green fluorescence) of JC1 was excited at 477 nm using a 500 nm dichroic mirror and fluoresced with a bandpass filter at 515–565 nm. Fluorescence measurements were performed at 30 s intervals using the same apparatus as Fura-2. As red fluorescence was eccentrically distributed around the nucleus and was often crescent-shaped, the ROI was defined as a hand-drawn polygonal area covering the region of high red fluorescence. The ratio of the mean fluorescence intensity (green/red) in the ROI was computed.

### 4.8. Cell Staining

To investigate the effects of melatonin on pharmacological PostC, dead cells in hippocampal slices after ischemic injury were visualized using propidium iodide and SYTOX -blue as membrane-impermeant fluorescent dyes for nuclear staining. Slices were incubated in aCSF containing 3 μM propidium iodide for 15 min starting 45 min prior to induction of the ischemic state. Slices were loaded with ischemic insult for 7.5 min, reperfused with aCSF for 20 min, then transferred to the incubation chamber. We compared the number of dead cells due to ischemic stress between slices treated with melatonin during ischemia-reperfusion (melatonin group) and slices reperfused with aCSF alone (control group). After incubation with aCSF for 3 h at 32 °C, slices were stained with 6 μM SYTOX-blue. Dead cells were examined under confocal microscopy (C2plus; Nikon, Tokyo, Japan). To detect dead cells prior to ischemia, propidium iodide was excited at 561 nm and the red fluorescence emission was bandpass filtered from 552 to 617 nm. SYTOX-blue was excited at 408 nm and the blue fluorescence emission was bandpass filtered from 417 to 477 nm. The number of dead cells showing only blue fluorescence in the CA1 region was counted.

### 4.9. Statistical Analysis

Data are presented as mean ± standard error of the mean. The Shapiro–Wilk test was used to test for normal distribution, and all results showed a normal distribution. Levene’s test was used to test for equal variances. Statistical testing involved Student’s *t*-test or Welch’s *t*-test, as appropriate. For studies employing multiple testing, we used one-way analysis of variance. Significant effects were further tested with a post-hoc multiple comparison test (Tukey–Kramer method or Games–Howell’s method, as appropriate). Significance was set at the level of *p* < 0.05.

## 5. Conclusions

Our findings demonstrate that melatonin-induced PostC inhibits the influx of Ca^2+^ into cytoplasm by decreasing NMDAR activity that is mediated through MTs. The depolarization of the mitochondrial inner membrane after an ischemia-reperfusion appears to play an important role in melatonin-induced PostC through low-conductance mPTP opening. Melatonin has the potential to become a future therapeutic agent for acute-phase ischemic strokes.

## Figures and Tables

**Figure 1 ijms-23-03822-f001:**
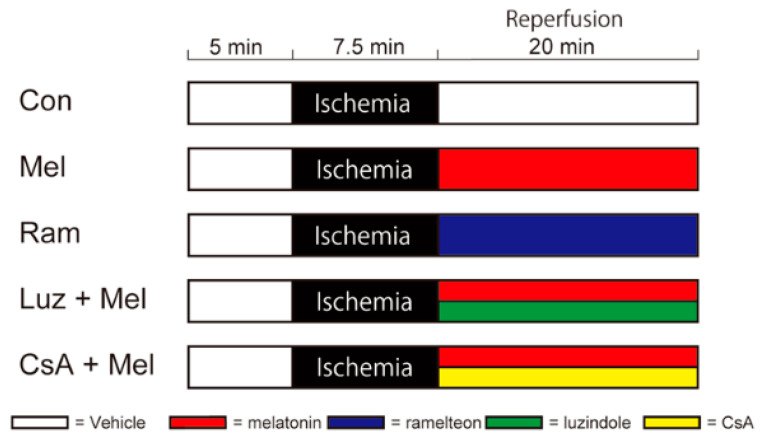
Diagram showing time schedules for ischemia and drug administration in each perfusion protocol. In each protocol, electrophysiological recording started collecting data 5 min before the start of ischemia and lasted up to 20 min after the reperfusion. The black band indicates the perfusion period during ischemia. White bands indicate perfusion with artificial cerebrospinal fluid. Red, blue, green, and yellow bands indicate administrations of melatonin, ramelteon, luzindole, and cyclosporine A in artificial cerebrospinal fluid, respectively. Con—control; Mel—melatonin; Ram—ramelteon; Luz—luzindole; CsA—cyclosporine A.

**Figure 2 ijms-23-03822-f002:**
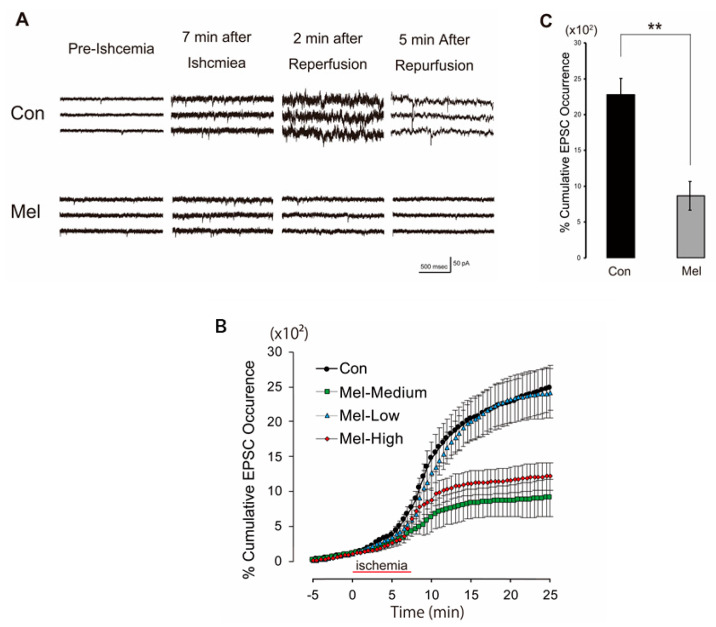
(**A**) Representative traces of spontaneous excitatory postsynaptic currents (sEPSCs) for control (upper) and melatonin (lower) groups (100 µM) during pre-ischemic, ischemic and reperfusion periods. In each trace, sEPSCs caused by synaptic glutamate releases are seen as transient downward deflections (inward currents). For both control and melatonin groups, occurrences of sEPSCs began to increase approximately 7 min after ischemic perfusion. In traces of the control group, an explosive increase in frequencies of sEPSCs were observed 2 min after reperfusion. In contrast, for the melatonin group, increased occurrences of sEPSCs quickly receded to pre-ischemic levels after reperfusion. (**B**) Time course of cumulative sEPSCs that occurred in control and melatonin-induced PostC groups. Cumulative sEPSCs that occurred were expressed as a percentage of the total number of sEPSCs occurring in the 5 min prior to ischemia under low (10 µM), medium (100 µM), and high (1 mM) concentrations of melatonin perfusion. In each group, the majority of sEPSCs occurred in the first 5 min after reperfusion. The timeline in the graph set to 0 min at the start for the ischemic load. In each group, the cumulative sEPSCs at 0 min of timeline were set as 100%. (**C**) Each vertical rectangle and error bar indicate percent cumulative sEPSCs that occurred at 20 min after onset of ischemic perfusion (12.5 min after reperfusion) and standard error of the mean (SEM), respectively. Asterisks indicate significant difference in *t*-test (** *p* < 0.01). Con—control; Mel—melatonin.

**Figure 3 ijms-23-03822-f003:**
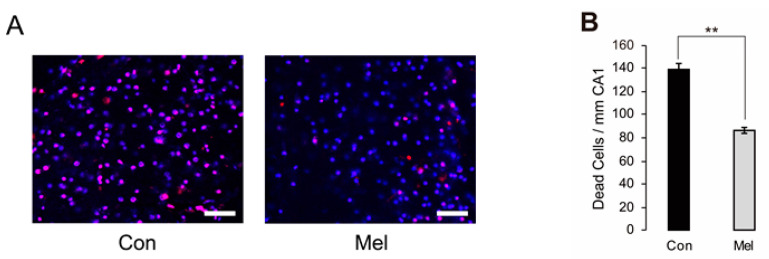
Comparison between control and melatonin-induced PostC groups for the number of dead neurons due to ischemic injury in the hippocampal CA1 region. (**A**) Microscopic view of the CA1 region shows the nuclei of dead cells in the control and melatonin-induced PostC groups, respectively. Magenta cells stained with both propidium iodide and SYTOX-blue were considered dead before electrophysiological recordings; blue cells stained with SYTOX-blue alone were considered dead due to ischemia-reperfusion injury. Scale bars = 50 μm. (**B**) The number of dead neurons per 1 mm of CA1 region. The number of dead neurons was significantly lower in the melatonin-induced PostC group according to *t*-testing (** *p* < 0.01). Con—control; Mel—melatonin.

**Figure 4 ijms-23-03822-f004:**
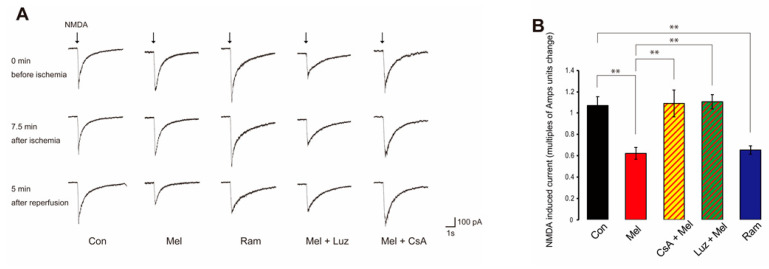
Effect of post-ischemic treatment and post-anoxic melatonin administration on N-methyl-D-aspartate (NMDA)-induced currents recorded from voltage-clamped hippocampal pyramidal neurons. (**A**) Typical traces of NMDA-induced currents prior to anoxia, at the end of anoxia, and after 5 min of anoxia in the Control, Mel, Ram, Mel + Luz, and Mel + CsA groups. Inward currents are represented by downward deflection. In the Mel group, NMDA-induced currents decreased and no change in waveform was seen after 5 min of anoxia. The control group showed no obvious change in NMDA-induced current. (**B**) Bar graph showing the change in mean peak amplitude of NMDA-induced current from 10 min to 20 min after anoxia in the control group, Mel group, Ram group, Mel + Luz group, and Mel + CsA group. Values are shown as currents in multiples of Amps units change relative to mean peak amplitude during the 5 min prior to anoxia. Asterisks indicate significant differences in Tukey–Kramer multiple comparisons test (** *p* < 0.01). NMDA—N-methyl-D-aspartate; Con—control; Mel—melatonin; Ram—ramelteon; Luz—luzindole; CsA—cyclosporine A.

**Figure 5 ijms-23-03822-f005:**
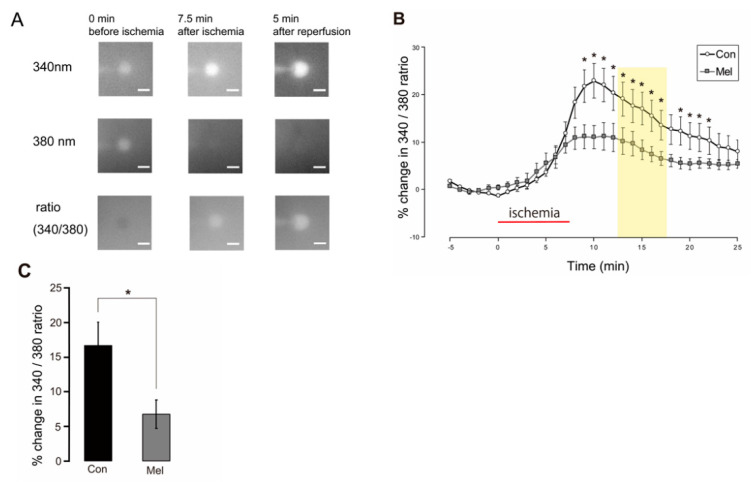
Effect of melatonin-induced PostC on cytosolic Ca^2+^ concentration after ischemia-reperfusion. (**A**) Representative microphotographs showing changes in Fura-2 emissions resulting from excitation at 340 and 380 nm for the control group. The elevation in the Fura-2 ratio (340/380 ratio) represents an increase in cytosolic Ca^2+^ concentration. Scale bars = 10 µm. (**B**) Course of changes in the Fura-2 ratio during pre-anoxic, anoxic, and reperfusion periods. Percentages are relative to the mean value observed during the 5 min of the pre-anoxic period. The red horizontal bar indicates the ischemic period. The timeline in the graph is set to 0 min at the start for the ischemic load. The increase in intracellular Ca^2+^ concentration after reperfusion is significantly inhibited by melatonin-induced PostC (*p* < 0.05). The yellow band represents the period used for statistical analysis. (**C**) Each vertical rectangle and error bar indicate percentage change in the Fura-2 ratio during 5–10 min after 7.5 min of ischemia (yellow band in (**B**)) and SEM, respectively. Asterisks indicate significant difference in *t*-test (* *p* < 0.05). Con—control; Mel—melatonin.

**Figure 6 ijms-23-03822-f006:**
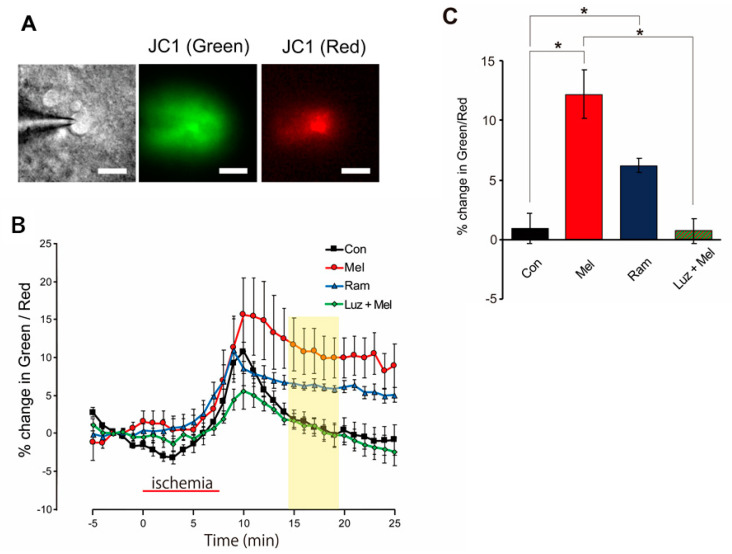
Changes in mitochondrial membrane potential as estimated from JC1 fluorescence during pre-ischemia, ischemic, and reperfusion periods. (**A**) Representative microphotographs of JC1 fluorescence in a slice of hippocampus: Left: infrared differential interference contrast image; Middle: green fluorescent image excited at 477 nm; Right: red fluorescent image excited at 548 nm. Scale bars = 10 µm. (**B**) Course of changes in mitochondrial membrane potential estimated with JC1 fluorescence during pre-anoxia, anoxia, and reperfusion periods. Percentages are relative to the mean value observed during the 5 min pre-anoxic period. The red horizontal bar indicates the ischemic period. The timeline in the graph set to 0 min at the start for the ischemic load. The yellow band represents the period used for statistical analysis. (**C**) Bar graph of percentage change in the JC1 green/red ratio, median data from the 7.5–12.5 min reperfusion period (yellow band in (**B**)). Asterisks indicate significant differences in Games–Howell multiple comparisons test (* *p* < 0.05). Con—control; Mel—melatonin; Ram—ramelteon; Luz—luzindole.

## Data Availability

The datasets of the current study are available upon request with no restrictions.

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
