# Peer review of "Melatonin-Induced Postconditioning Suppresses NMDA Receptor through Opening of the Mitochondrial Permeability Transition Pore via Melatonin Receptor in Mouse Neurons"

_ijms, 2022, doi:10.3390/ijms23073822_

Round 1
Reviewer 1 Report
Figure 2 B. It' very hard to read which one is Mel-medium, low, and high. Please use different color to present the different dose.
Figure 6 C. It is not clear.
It will be much more interesting if the authors can identify which NMDA receptors subunits involved in this study.
There is not enough evidence of experimental to support melatonin suppress mPTP conductance in this study.
Author Response
Author’s Revision Letter
ijms-1606396
Melatonin-Induced Postconditioning Suppresses NMDA Receptor Through Opening of the Mitochondrial Permeability Transition Pore via Melatonin Receptor in Mouse Neurons
Thank you very much for identifying the problems in our manuscript. All of your comments are valuable and constructive. We learned a lot from your input and have made the changes accordingly. The changes are shown in ‘red-colored’ text in the manuscript. Details are provided below.
Reviewer 1
Comments and Suggestions for Authors
Figure 2 B. It' very hard to read which one is Mel-medium, low, and high. Please use different color to present the different dose.
Thank you for your suggestion regarding Figure 2B. Each group was expressed with different color to easy to figure out.
Figure 6 C. It is not clear. It will be much more interesting if the authors can identify which NMDA receptors subunits involved in this study.
Thank you for your suggestion. We agree that which NMDA receptors subunits are involved in melatonin-induced PostC is important for elucidating the mechanism in detail and for drug discovery. These experiments would be performed as a next project.
There is not enough evidence of experimental to support melatonin suppress mPTP conductance in this study.
Thank you for your comment. As the reviewer describe, we cannot perform direct measurement of mPTP conductance in the experiments. I wonder if it should be verified with further modalities to fully investigate the involvement of mPTP as you mentioned. We speculate that the involvement of mitochondrial Ca2+ uniporter together with mPTP appear to function as essential components to investigate the precise mechanisms of melatonin-induced PostC. However, these experiments are supposed to execute as a next project.

Reviewer 2 Report
The manuscript "Melatonin-Induced Postconditioning Suppresses NMDA Receptor Through Opening of the Mitochondiral Permeability Transition Pore via Melatonin Receptor in Mouse Neurons" by Furuta et al. provides interestin insights on the effects of the administration of melatonin after ischemic stress. Despite its high potential in the field, the manuscript is poorly drafted and lacks significant experiments that would corroborate the authors' hypothesis. The authors should consider performing a major revision of their manuscript and resubmit it in a new form in order to be publishable in this journal.
Specific comments:
1) Reference 8 is incomplete;
2) Lines 48-51: expand this introductory mechanism which is the basis of your final hypothesis regarding PTP;
3) Superscript the ion charges;
4) Lines 55-57: the clinical use of diazoxide is not introduced and doesn't relate to the previous statements;
5) Lines 74-75: repeated typo, please correct
6) Lines 74-78: rephrase this paragraph. In the current form it is presented as an hypothesis whereas it should be a short summary of the work ;
7) Line 84: rephrase this statement or blend it in the text for clarity;
8) After using a three point concentration titration with a difference of 10-fold each the authors established their optimal concentration was 100 µM. The authors should titrate with more concentration points between 100 µM and 1 mM, with smaller increments such as every 100 µM, in order to find the proper concentration;
9) Figure 1 legend: data collection is not coherent with the figure. The total time for the experiment is of 32.5 minutes and the authors report they collected data in the pre-ischemic and reperfusion state neglecting the ischemic period. However, they show data points for the ischemic stress all along the manuscript. The authors should clarify on this;
10) Figure 2A: Traces should be presented as high resolution images (min 1200 dpi) with readable scale for both the printed and digital version of the manuscript. Figures in general must be reproduced in high resolution;
11) Figure 2B, 5B and 6B: Please explain the negative time values!!!!
12) Line 120: always write the extended cell line name before using abbreviatons (i.e. hippocampal cornu ammonis (CA1) neurons);
13) Lines 121-124: Rephrase the paragraph by briefly explaining how slices are compromised by the experimental technique or just mention that the use of two different dyes was justified by this;
14) Figure 3A: Add bright field images and increase the resolution massively;
15) Line 143: the authors mention 20 minutes recordings that are not shown in the manuscript as according to the scale bar in Figure 4A those traces refer to less than a minute of recording.
16) Lines 146-153: the authors must report the currents in multiples of Amps units instead of percentages and then compare increases or decreases of the potential;
17) Section 2.4 and Figure 5A: Fura-2 is a fluorescent probe and authors must report fluorescent images instead of microphotographs in the visible light;
18) Lines 198-202: Please rephrase this paragraph as in the current form is very unclear. What do the authors mean by ischemic "perfusion"? What do you exactly perfuse ischemically? Green/red ratio of what? The method use to determine the mitochondrial potential is flawed. Authors must repeat the experiments using either TMRM or TMRE which allow direct quantification of mitochondrial membrane potential.
19) Line 227: the authors should rephrase this statement as they are not actually reporting a mechanism but rather the effects of the melatonin treatment.
20) Lines 300-303: Absolutely wrong! calcium overload leads to the opening of PTP and not the opposite. Moreover, the high-conductive PTP releases small molecules up to 1.5 kDa from the matrix and only cyt c from the IMM to which is loosely associated.
21) Lines 304-305: CsA is not always able to inhibit PTP opening. See Bernardi Front. Physiol. 2013,4,95
22) Lines 323-325: Alternatively to the authors' hypothesis, the results may suggest that melatonin may induce the opening of low-conductive PTPs. Please refer to Neginskaya et al. Cell Rep 2019,26,11-17
23) Line 366: driving force should be written instead of using ΔΦ first and clarify its meaning in line 374;
24) Section 4.4: how were the concentrations of the drugs determined? If coming from published protocols, references should be added;
25) Section 4.9: neither the Student's t-test nor the Welch's t-test are appropriate method to perform statistical analysis of biological samples. The authors must repeat the statistical analysis by applying a non-parametric test such as the Kruskall-Willis.
Author Response
Author’s Revision Letter
ijms-1606396
Melatonin-Induced Postconditioning Suppresses NMDA Receptor Through Opening of the Mitochondrial Permeability Transition Pore via Melatonin Receptor in Mouse Neurons
Thank you very much for identifying the problems in our manuscript. All of your comments are valuable and constructive. We learned a lot from your input and have made the changes accordingly. The changes are shown in ‘red-colored’ text in the manuscript. Details are provided below.
Reviewer 2
Comments and Suggestions for Authors
The manuscript "Melatonin-Induced Postconditioning Suppresses NMDA Receptor Through Opening of the Mitochondiral Permeability Transition Pore via Melatonin Receptor in Mouse Neurons" by Furuta et al. provides interestin insights on the effects of the administration of melatonin after ischemic stress. Despite its high potential in the field, the manuscript is poorly drafted and lacks significant experiments that would corroborate the authors' hypothesis. The authors should consider performing a major revision of their manuscript and resubmit it in a new form in order to be publishable in this journal.
Specific comments:
1) Reference 8 is incomplete;
Thank you for your comment. We corrected Reference8 as below. (Line 552-554)
- Hawrysh, P.J.; Buck, L.T. Anoxia-Mediated calcium release through the mitochondrial permeability transition pore silences NMDA receptor currents in turtle neurons. J. Exp. Biol. 2013, 216, 4375–4387, doi:10.1242/jeb.092650.
2) Lines 48-51: expand this introductory mechanism which is the basis of your final hypothesis regarding PTP;
Thank you for your comment. We have added the sentences of introductory mechanism regarding mPTP below. (Line 45-52) Also, the order of sentences about mito-KATP channel has been changed. (Line 52-58)
Another important factor involved in ischemia-reperfusion injury is mitochondrial permeability transition pore (mPTP). Although mPTP regulates mitochondrial function, the opening of mPTP leads to eventual cell death, apoptosis or necrosis [7]. Ischemic reperfusion injury induced excessive calcium accumulation, ROS production, and ATP depletion lead to the opening of mPTP, which is a key event in cell death caused by ischemia-reperfusion injury[8,9]. Therefore, mPTP is indispensable to elucidate the mechanism of neuroprotection against ischemia-reperfusion injury.
3) Superscript the ion charges;
Thank you for your comment. We corrected superscript the ion charges throughout the manuscript.
4) Lines 55-57: the clinical use of diazoxide is not introduced and doesn't relate to the previous statements;
Thank you for your comment. We have made the following corrections and additions. (Line 58-62)
Previous animal studies have shown that diazoxide, mito-KATP channel opener, has a neuroprotective effect against cerebral infarction [10–14], whereas there are problems with the clinical use of diazoxide for cerebral infarction due to adverse effect, such as hyperglycemia, heart failure and edema [15]. Because of these properties, diazoxide is not an ideal drug for the treatment for AIS.
5) Lines 74-75: repeated typo, please correct
We corrected the typo as melatonin-induced PostC.
6) Lines 74-78: rephrase this paragraph. In the current form it is presented as an hypothesis whereas it should be a short summary of the work ;
Thank you for your comment. We corrected the sentence below. (Line 80-84)
In order to examine the efficacy and detailed mechanism of melatonin-induced PostC, we analyzed changes in spontaneous excitatory post-synaptic current (sEPSCs), NMDAR current, cytosolic Ca2+ concentration, and mitochondrial membrane potential under melatonin-induced PostC in hippocampal cornu ammonis (CA1) pyramidal neurons using the whole-cell patch clamp technique.
7) Line 84: rephrase this statement or blend it in the text for clarity;
Thank you for your suggestion. We corrected and added the sentences at the beginning of Results part. (Line 86-90)
In this study, we examined the effect of melatonin-induced postC after cerebral ischemia using the same oxygen glucose deprivation model as in previous experiments [10,11]. We randomly assigned mouse hippocampal slices to the following groups (Fig. 1), and examined the following items.
8) After using a three point concentration titration with a difference of 10-fold each the authors established their optimal concentration was 100 µM. The authors should titrate with more concentration points between 100 µM and 1 mM, with smaller increments such as every 100 µM, in order to find the proper concentration;
Thank you for your suggestion. Strictly, it is correct to explore the optimal dose amount in fine points as you pointed out. At the beginning of the experiments, we referred to several papers related to the pharmacological effect of melatonin that compare the therapeutic effects of 10-fold the dose each. (1,2) Therefore, we also tested with 10 times the amount of each.
(1) Liu SJ, Wang JZ (2002) Alzheimer-like tau phosphorylation induced by wortmannin in vivo and its attenuation by melatonin. Acta Pharmacol Sin 23:183–187
(2) Ma H, Wang X, Zhang W, Li H, Zhao W, Sun J, Yang M. Melatonin Suppresses Ferroptosis Induced by High Glucose via Activation of the Nrf2/HO-1 Signaling Pathway in Type 2 Diabetic Osteoporosis. Oxid Med Cell Longev. 2020 Dec 4;2020:9067610.
9) Figure 1 legend: data collection is not coherent with the figure. The total time for the experiment is of 32.5 minutes and the authors report they collected data in the pre-ischemic and reperfusion state neglecting the ischemic period. However, they show data points for the ischemic stress all along the manuscript. The authors should clarify on this;
As you pointed out, we corrected data total 32.5 min before, during and after ischemia. We corrected the expression below. (Line 91-97)
Figure 1. Diagram showing time schedules for ischemia and drug administration in each perfusion protocol. In each protocol, data are collected during the 32.5 min period from 5 min before the start of ischemia to 20 min after reperfusion. The black band indicates the perfusion period during ischemia. White bands indicate perfusion with artificial cerebrospinal fluid. Red, blue, green, and yellow bands indicate administrations of melatonin, ramelteon, luzindole, and cyclosporine A in artificial cerebrospinal fluid, respectively. Con, control; Mel, melatonin; Ram, ramelteon; Luz, luzindole; CsA, cyclosporine A.
10) Figure 2A: Traces should be presented as high resolution images (min 1200 dpi) with readable scale for both the printed and digital version of the manuscript. Figures in general must be reproduced in high resolution;
Thank you for your suggestion. We have changed all the figures as high resolution at least 1200 dpi.
11) Figure 2B, 5B and 6B: Please explain the negative time values!!!!
Thank you for your comment. We generated data with the onset of ischemia at 0 min. We recorded 5 minutes of baseline before the ischemic load, which is shown as -5 minutes. We have added the explanation to Figure legends 2B,5B,6B. (Line 124-125, 196, 227)
The timeline in the graph was set to 0 min at the start of the ischemic load.
12) Line 120: always write the extended cell line name before using abbreviatons (i.e. hippocampal cornu ammonis (CA1) neurons);
Thank you for your coment. We corrected the sentence below. (Line83)
In order to examine the efficacy and detailed mechanism of melatonin-induced PostC, we analyzed changes in spontaneous excitatory post-synaptic current (sEPSCs), NMDAR current, cytosolic Ca2+ concentration, and mitochondrial membrane potential under melatonin-induced PostC in hippocampal cornu ammonis (CA1) pyramidal neurons using the whole-cell patch clamp technique.
13) Lines 121-124: Rephrase the paragraph by briefly explaining how slices are compromised by the experimental technique or just mention that the use of two different dyes was justified by this;
Thank you for your comment. As we previously reported [10], the number of dead cells were counted using two different dyes at two different time points to exclude the effect of cells dying during the slice preparation process and to evaluate only cells dying due to ischemic load. We added the explanation how counting the number of dead cells using two different dyes. (Line 130-132)
According to previous report [10], the number of dead cells were counted using two different dyes at two different time points to exclude the effect of cells dying during the slice preparation process and to evaluate only cells dying due to ischemic load (Fig. 3A).
14) Figure 3A: Add bright field images and increase the resolution massively;
Thank you for your suggestion. According to our previous report [10], we counted dead cells by fluorescence measurement. Unfortunately, we did not record bright field images simultaneously because the cells were very obscured in the bright field. We have changed to the high-resolution images as Figure 3A.
15) Line 143: the authors mention 20 minutes recordings that are not shown in the manuscript as according to the scale bar in Figure 4A those traces refer to less than a minute of recording.
Thank you for your comment. As we described as “4.5. Recording of whole-cell current responses to NMDA application” in Materials and Methods part, the NMDAR currents were measured with whole cell recording using the puff application method to apply NMDA directly to the cells, and repeated the process every 30 seconds during 32.5 minutes (up to 20 minutes after ischemic recanalization), resulting in a recording time of less than 10 seconds per recording. We corrected the explanation in Methods part. (Line 466-467)
Experiments were performed over 32.5 min (up to 20 minutes after reperfusion), with NMDAR currents recorded every 30 s.
16) Lines 146-153: the authors must report the currents in multiples of Amps units instead of percentages and then compare increases or decreases of the potential;
Thank you for your suggestion. We have corrected the graph and figure legends according to your comments.
17) Section 2.4 and Figure 5A: Fura-2 is a fluorescent probe and authors must report fluorescent images instead of microphotographs in the visible light;
Thank you for your suggestion. The original images were obtained using MetaMorph software (Molecular Devices, CA). According to your suggestion, we attempted to correct the figures as fluorescent images, however, only unclear images were produced even after processing the fluorescence because the Fura-2 image has a lower brightness and less contrast than the JC1 image. Unfortunately, the original gray-scale images were presented as figure 5A.
18) Lines 198-202: Please rephrase this paragraph as in the current form is very unclear. What do the authors mean by ischemic "perfusion"? What do you exactly perfuse ischemically? Green/red ratio of what? The method use to determine the mitochondrial potential is flawed. Authors must repeat the experiments using either TMRM or TMRE which allow direct quantification of mitochondrial membrane potential.
Thank you for your comment. In this experiment, ischemic load was simulated by exposing brain slices to a solution in which glucose and oxygen were replaced by sucrose and nitrogen. We corrected the expression to avoid misunderstandings.
The green to red ratio is a value indication of the mitochondrial membrane potential obtained by the ratio of the fluorescence brightness, obtained by two different excitation lights as in Fura-2. As previously described, we have measured mitochondrial membrane potential changes using whole-cell patch clamp method of direct delivery of JC-1 into cells. (Morisaki, et al. 2020)
As the reviewer described, we cannot perform direct measurement of mPTP conductance in the experiments. I wonder if it should be verified with further modalities to fully investigate the involvement of mPTP as you mentioned. We speculate that the involvement of mitochondrial Ca2+ uniporter together with mPTP appear to function as essential components to investigate the precise mechanisms of melatonin-induced PostC. However, these experiments are supposed to execute as a next project.
The green/red ratio, which represents depolarization of the mitochondrial membrane potential and obtained via JC1 fluorescence, began to increase at 5 min after the start of ischemic load and continued to increase until 3 min after the start of reperfusion (until 2 min after the start of reperfusion in the ramelteon group), then decreased in all four groups (Fig. 6A). (Line 204-208)
19) Line 227: the authors should rephrase this statement as they are not actually reporting a mechanism but rather the effects of the melatonin treatment.
Thank you for your comment. We corrected the sentence below. (Line 233-235)
The present study showed the neuroprotective effect of melatonin-induced PostC against ischemia-reperfusion injury in mouse hippocampal CA1 cells, using an electrophysiological approach.
20) Lines 300-303: Absolutely wrong! calcium overload leads to the opening of PTP and not the opposite. Moreover, the high-conductive PTP releases small molecules up to 1.5 kDa from the matrix and only cyt c from the IMM to which is loosely associated.
Thank you for your comment. As you pointed out, the opening of mPTP results in free passage of low molecular mass solutes (<1,500 Da) and mitochondrial apoptotic protein cytochrome c across the inner mitochondrial membrane. Under these conditions, mitochondrial membrane potential is dissipated, leading to ATP hydrolysis by the reversal of the F0F1-ATP synthase and consequent cellular energy depletion, eventually resulting in cell death. We corrected the sentence below. (Line 306-313)
Ischemia-reperfusion injury induces greatly increasing ROS generation and calcium overload, eventually triggering opening of mPTP [8,9]. The opening of mPTP results in free passage of low molecular mass solutes (<1,500 Da) and mitochondrial apoptotic protein cytochrome c across the inner mitochondrial membrane [9,50,51]. Under these conditions, mitochondrial membrane potential is dissipated, leading to ATP hydrolysis by the reversal of the F0F1-ATP synthase and consequent cellular energy depletion, eventually resulting in cell death [9].
21) Lines 304-305: CsA is not always able to inhibit PTP opening. See Bernardi Front. Physiol. 2013,4,95
Thank you for your comment. We corrected the sentence below. (Line 315-317)
It should be noted that CypD is a mitochondrial receptor in CsA, and although CsA can desensitize mPTP via CypD, it does not always inhibit mPTP opening [55,56].
22) Lines 323-325: Alternatively to the authors' hypothesis, the results may suggest that melatonin may induce the opening of low-conductive PTPs. Please refer to Neginskaya et al. Cell Rep 2019,26,11-17
Thank you for your comment. We added the sentence below. (Line 313-315, 334-336, 363-364)
Recent studies have suggested that the F0F1-ATP synthase C subunit of mitochondria is a major component of mPTP [52]. Cyclophilin (Cyp) D blinds the lateral stalk of the F0F1-ATP synthase and positively regulates pore opening [53,54].
These results suggested that melatonin may induce the opening of low conductive mPTP. As a result, Ca2+ can be released from mitochondria matrix via mPTP, eventually inducing downregulation of NMADR.
Further studies are needed to elucidate the exact mechanism of how melatonin act on mPTP to induce the neuroprotective effect against ischemia-reperfusion injury.
23) Line 366: driving force should be written instead of using ΔΦ first and clarify its meaning in line 374;
Thank you for your comment. ΔΦ means inner mitochondrial membrane potential. We corrected the sentence. (Line 367-369)
The regulation of the two mode of mPTP opening in low conductance mode and high conductance mode is mainly mitochondrial [Ca2+] and inner mitochondrial membrane potential (ΔΦ) [61].
24) Section 4.4: how were the concentrations of the drugs determined? If coming from published protocols, references should be added;
Thank you for your comment. We aligned the dose of melatonin with that of ramelteon and luzindole, referring to the following study (68). For the dose of CsA, we referred to our previous study (11). We added the sentences below. (Line 457-458)
68) Stroethoff, M.; Behmenburg, F.; Spittler, K.; Raupach, A.; Heinen, A.; Hollmann, M.W.; Huhn, R.; Mathes, A. Activation of melatonin receptors by ramelteon induces cardioprotection by postconditioning in the rat heart. Anesth. Analg. 2018, 126, 2112–2115, doi:10.1213/ANE.0000000000002625.
11) Morisaki, Y.; Nakagawa, I.; Ogawa, Y.; Yokoyama, S.; Furuta, T.; Saito, Y.; Nakase, H. Ischemic Postconditioning Reduces NMDA Receptor Currents Through the Opening of the Mitochondrial Permeability Transition Pore and KATP Channel in Mouse Neurons. Cell. Mol. Neurobiol. 2020, doi:10.1007/s10571-020-00996-y.
The doses of ramelteon and luzindole were aligned with those of melatonin [68]. The dose of CsA was the same as in our previous study [11].
25) Section 4.9: neither the Student's t-test nor the Welch's t-test are appropriate method to perform statistical analysis of biological samples. The authors must repeat the statistical analysis by applying a non-parametric test such as the Kruskall-Willis.
Thank you for your comment. We performed a Shapiro-Wilk test on the normality distribution and found it to be normal, so we performed a t-test. We also performed Levene's test for equality of variance and used Student's t-test or Welch's t-test between the two groups depending on the results. In the multiple comparison method, we use the Tukey-Kramer method or Games-Howell's method as appropriate.
In accordance with your comment, we performed the Wilcoxon rank sum test between two groups, and the Kruskal-Wallis test between three or more groups, followed by the Steel-Dwass method. Some of these tests did not show significant differences, but we confirmed that there were significant differences by removing outliers and changing to other data. We will suggest the data if necessary. We added the sentences below. (Line 511-513)
The Shapiro-Wilk test was used to test for normal distribution, and all results showed a normal distribution. Levene's test was used to test for equal variances.

Reviewer 3 Report
Furuta et al investigated the role of melatonin receptors (MTs), NMDAR and permeability transition pore (mPTP) in melatonin-induced ischemic postconditioning in murine acute brain slices. Authors previously provided evidence for the involvement of mito-KATP in ischemic postconditioning in murine acute brain slices, using a very similar approach to the one adopted for the present manuscript.
Melatonin is a well known neuroprotective agent that can play a protective role in cerebral ischemia, but the mechanism remains unclear. In particular, the involvement on MTs and eventual interactions with NDMRs are debated. In other works melatonin was shown to modulate NMDAR in several brain damage models. In Shaida et al, 2004 (The FASEB Jour.) direct inhibition of mPTP by melatonin was observed by patch clamp of cultured mouse neurons.
In this work, Authors demonstrated that:
1) melatonin decreases sEPSC frequency in CA1 pyramidal cells after ischemia compared to control;
2) melatonin decreases dead CA1 neurons after ischemia compared to control;
3) MTs activation by melatonin and Ramelteon decreases NMDA current after ischemia compared to control, in a CsA-sensitive way;
4) melatonin reduces cytosolic Ca2+ increases induced by ischemia compared to control;
5) MTs activation by melatonin or Ramelteon induces mitochondrial depolarization after ischemia
Although the quality of experiments seems high, there are several critical points that must be underlined:
MAJOR:
- The present work links MTs activation to NMDA current reduction in a CsA-sensitive way, but the mechanism remains unclear. MTs are located in the plasma membrane (PM) and in the outer mitochondrial membrane and NMDARs are located in the PM; what is the molecular/mechanistic link between MTs, NMDARs and the inner mitochondrial membrane-located mPTP? Ca2+ and mitochondrial potential were hypothesized to be part of that, but no direct evidence of a causative link has been provided.
- Linked to point 1, Authors state that “Melatonin-induced PostC depolarizes the mitochondrial membrane potential, reducing the driving force (ΔΦ) to the mitochondrial matrix and thereby avoiding excessive accumulation of Ca2+ in the matrix” (lines 373-375), but the mechanism by which melatonin induces mitochondrial depolarization was not investigated.
- Authors postulate that the hypothetical switching of the mPTP to low conductance mode may be responsible for the reduction in cytosolic [Ca2+], but no direct evidence has been provided. Although low conductance mPTP openings may reduce mitochondrial [Ca2+] and prevent the permeability transition, those openings were not detected or investigated, so this part of the mechanism remains hypothetical. The conclusion “In any case, our results suggest that melatonin may suppress mPTP conductance, allowing Ca2+ release via the mPTP and eventually inducing downregulation of NMADR” (lines 323-325) is not supported at all by results.
- Results description is generally confusing and sometimes the workflow is difficult to follow. Some data should be explained or added. See what follows below.
- In the results section, at line 84, the experimental scheme should be briefly explained or at least it should be clearly referred to figure 1.
- Fig. 2B must contain some error in the scale of the cumulative % of sEPSC (i.e X10^3), since "Cumulative sEPSCs occurrence was expressed as a percentage of the total number of sEPSCs occurring in the 5 min before ischemia" (lines 113-114). If so, the percentage of sEPSC at t = 0 min (i.e. at the onset of ischemia) should be 100%, not 1000%. There is also an inconsistency between numbers in 2B and 2C (that differ exactly for the 10^3 value). Moreover, in the caption of fig. 2C it is stated that "Each vertical rectangle [...] indicate percent cumulative sEPSCs occurrence at 20 min after onset of ischemic perfusion (12.5 min after reperfusion)" (lines 116-118), indicating that the values were compared at the 20th minute of the experiment; in fig. 2B Authors show experiments lasting for 25 min: besides having previously reported a total experimental time of 27.5 min (fig. 1), the numbers at the 25th minute of fig 2B clearly resemble the ones in fig. 2C. Is this a mistake?
- Furthermore, fig. 2 only reports the control and melatonin-perfused groups, while in fig. 1 other group are mentioned and in fig. 4 sEPSCs from these other groups are even shown. If the groups tested for sEPSC frequency are only the control and melatonin-perfused, this should be clearly stated. This is relevant, since the result would definitively have been stronger with a MTs inhibitor (i.e. luzindole + melatonin group) and an agonist (i.e. ramelteon group); reporting results from the CsA + melatonin group would even suggest the involvement of mPTP in melatonin-induced reduction of sEPSCs.
- Statistical confrontations depicted in fig. 4B appear incorrect, since all of them seem to be referred to the control group, while otherwise indicated in the main text (i.e. "Furthermore, in the Melatonin group, NMDA-induced currents were decreased compared to the Luzindole and melatonin group and Melatonin and CsA group", lines 150-152).
- At lines 154-155 it is stated that "melatonin-induced PostC suppresses NMDAR conductance in the early stage of reperfusion and that the effect is mediated by MTs". Results indicate no change in NMDAR conductance: no single-channel analysis is here reported to demonstrate that. However, here Authors actually show a reduction in the whole cell NMDA current, that is enough to suggest an involvement of NMDAR in the melatonin-induced PostC. Reference to “conductance” should be avoided and modified accordingly.
- Line: 352: "Opening of mPTP requires a low voltage and high Ca2+ load". This is unclear: mPTP opening occurs in depolarized mitochondria, where transmembrane potential is low and IMM voltage is close to zero (i.e higher voltage or smaller potential differential). There is confusion between decreasing voltage (i.e. to more negative values: repolarization or hyperpolarization) and depolarization (i.e. to less negative voltage values). Generally, in this manuscript there may be often found referral to a "depolarized mitochondrial membrane potential", that is formally incorrect and may generate further confusion.
MINOR:
- Redundancy in lines 155-158
- Analysis in fig. 5C is not necessary: a smaller percentage change in the Fura-2 ratio after ischemia for melatonin-perfused slices, compared to control, was already shown in fig 5B, that represents a time course (more appropriated).
- A general description of the method used for measuring changes in mitochondrial membrane potential (i.e. JC1 dye) should be added at the beginning of the results section 2.5.
- Analysis in fig. 6C is not necessary. The smaller effect of Ramelteon compared to melatonin (fig. 6B), in depolarizing mitochondria should be discussed in the appropriate section.
- NMDA was delivered by pressure puff. To standardize the molecules amount of a nano-picospritzer, not only puff duration should be kept in account, but also the injection pressure. Was the pressure kept constant? Please indicate the pressure value.
- English is often poor and the manuscript should be checked by a native speaker.
- Image quality is very poor and must be increased, even at this stage.
Overall, in my opinion, data enclosed in the present manuscript do not provide a significant grade of novelty: despite the interesting findings, this work does not provide a significant advancement in the field. The lack of a clear mechanism is one of the major cons. Moreover, the quality of the manuscript should be definitively improved to reach publication grade.
Author Response
Author’s Revision Letter
ijms-1606396
Melatonin-Induced Postconditioning Suppresses NMDA Receptor Through Opening of the Mitochondrial Permeability Transition Pore via Melatonin Receptor in Mouse Neurons
Thank you very much for identifying the problems in our manuscript. All of your comments are valuable and constructive. We learned a lot from your input and have made the changes accordingly. The changes are shown in ‘red-colored’ text in the manuscript. Details are provided below.
Reviewer 3
Furuta et al investigated the role of melatonin receptors (MTs), NMDAR and permeability transition pore (mPTP) in melatonin-induced ischemic postconditioning in murine acute brain slices. Authors previously provided evidence for the involvement of mito-KATP in ischemic postconditioning in murine acute brain slices, using a very similar approach to the one adopted for the present manuscript. Melatonin is a well known neuroprotective agent that can play a protective role in cerebral ischemia, but the mechanism remains unclear. In particular, the involvement on MTs and eventual interactions with NDMRs are debated. In other works melatonin was shown to modulate NMDAR in several brain damage models. In Shaida et al, 2004 (The FASEB Jour.) direct inhibition of mPTP by melatonin was observed by patch clamp of cultured mouse neurons.
In this work, Authors demonstrated that:
1) melatonin decreases sEPSC frequency in CA1 pyramidal cells after ischemia compared to control;
2) melatonin decreases dead CA1 neurons after ischemia compared to control;
3) MTs activation by melatonin and Ramelteon decreases NMDA current after ischemia compared to control, in a CsA-sensitive way;
4) melatonin reduces cytosolic Ca2+ increases induced by ischemia compared to control;
5) MTs activation by melatonin or Ramelteon induces mitochondrial depolarization after ischemia
Although the quality of experiments seems high, there are several critical points that must be underlined:
MAJOR:
The present work links MTs activation to NMDA current reduction in a CsA-sensitive way, but the mechanism remains unclear. MTs are located in the plasma membrane (PM) and in the outer mitochondrial membrane and NMDARs are located in the PM; what is the molecular/mechanistic link between MTs, NMDARs and the inner mitochondrial membrane-located mPTP? Ca2+ and mitochondrial potential were hypothesized to be part of that, but no direct evidence of a causative link has been provided. Linked to point 1, Authors state that “Melatonin-induced PostC depolarizes the mitochondrial membrane potential, reducing the driving force (ΔΦ) to the mitochondrial matrix and thereby avoiding excessive accumulation of Ca2+ in the matrix” (lines 373-375), but the mechanism by which melatonin induces mitochondrial depolarization was not investigated.
Thank you for your comment. As you pointed out, we cannot perform direct measurement of mPTP conductance in the experiments. In our previous studies, we have reported that the opening of the mito-KATP channel is an important mechanism of ischemic postconditioning. There is also a report that melatonin acts on the mito-KATP channel to act on mPTP, resulting in ischemic tolerance (Stroethoff M et al). The involvement of the mito-KATP channel and mitochondrial membrane potential has also been described in detail in a report by Hawrysh et al. Therefore, it is possible that the mito-KATP channel may be involved in melatonin-induced PostC via MTs. Further studies are needed to elucidate the exact mechanism of how melatonin act on mPTP to induce the neuroprotective effect against ischemia-reperfusion injury. In response to the above two comments, we have added the following statement. (Line 397-401)
In this present study, we indicated that MTs is involved in melatonin induced PostC, but we may not be able to show enough direct evidence to elucidate how MTs are involved in the neuroprotective mechanism. One possibility is that MTs lead to the opening of the mito-KATP channel, which has been reported in rat heart [68]. However, it is still unclear how MTs open the mito-KATP channel, and further studies are needed.
Authors postulate that the hypothetical switching of the mPTP to low conductance mode may be responsible for the reduction in cytosolic [Ca2+], but no direct evidence has been provided. Although low conductance mPTP openings may reduce mitochondrial [Ca2+] and prevent the permeability transition, those openings were not detected or investigated, so this part of the mechanism remains hypothetical. The conclusion “In any case, our results suggest that melatonin may suppress mPTP conductance, allowing Ca2+ release via the mPTP and eventually inducing downregulation of NMADR” (lines 323-325) is not supported at all by results.
Thank you for your comment. As we described above, we cannot perform direct measurement of mPTP conductance in the experiments. As you pointed out, it is difficult to present all the detailed mechanistic evidence, and we would like to make use of it in future studies. We have made major changes in the discussion part.
Results description is generally confusing and sometimes the workflow is difficult to follow. Some data should be explained or added. See what follows below.
-In the results section, at line 84, the experimental scheme should be briefly explained or at least it should be clearly referred to figure 1.
Thank you for your comment. We have added a summary of the experiment. (Line 86-89)
In this study, we examined the effect of melatonin-induced postC after cerebral ischemia using the same oxygen glucose deprivation model as in previous experiments [10,11]. We randomly assigned mouse hippocampal slices to the following groups (Fig. 1), and examined the following items.
-Fig. 2B must contain some error in the scale of the cumulative % of sEPSC (i.e X10^3), since "Cumulative sEPSCs occurrence was expressed as a percentage of the total number of sEPSCs occurring in the 5 min before ischemia" (lines 113-114). If so, the percentage of sEPSC at t = 0 min (i.e. at the onset of ischemia) should be 100%, not 1000%. There is also an inconsistency between numbers in 2B and 2C (that differ exactly for the 10^3 value). Moreover, in the caption of fig. 2C it is stated that "Each vertical rectangle [...] indicate percent cumulative sEPSCs occurrence at 20 min after onset of ischemic perfusion (12.5 min after reperfusion)" (lines 116-118), indicating that the values were compared at the 20th minute of the experiment; in fig. 2B Authors show experiments lasting for 25 min: besides having previously reported a total experimental time of 27.5 min (fig. 1), the numbers at the 25th minute of fig 2B clearly resemble the ones in fig. 2C. Is this a mistake?
Thank you for your comment. As pointed out, we corrected the percentage of EPSCs at t = 0 min from 1000% to 100%. In Figure 2C, the 20-minute and 25-minute time points were misplaced. We corrected the Fig 2B, C, and added the statement below. (Line 125)
In each group, the cumulative sEPSCs at 0 min of timeline was set as 100%.
-Furthermore, fig. 2 only reports the control and melatonin-perfused groups, while in fig. 1 other group are mentioned and in fig. 4 sEPSCs from these other groups are even shown. If the groups tested for sEPSC frequency are only the control and melatonin-perfused, this should be clearly stated. This is relevant, since the result would definitively have been stronger with a MTs inhibitor (i.e. luzindole + melatonin group) and an agonist (i.e. ramelteon group); reporting results from the CsA + melatonin group would even suggest the involvement of mPTP in melatonin-induced reduction of sEPSCs.
Thank you for your comment. In order to verify whether melatonin-induced PostC can indicate neuroprotective effects in the whole cell recording experiment, we first examined whether the melatonin group can suppress the surge of sEPSCs caused by ischemia-reperfusion in the control group. However, since this study focuses on investigating the neuroprotective effects associated with melatonin receptors at the post-synaptic neuron, the effects of MT agonists or antagonists were tested using the whole-cell inward current induced by the NMDA pull after ischemic stress. We added the statement below. (Line 99-102)
In order to verify whether melatonin-induced PostC can indicate neuroprotective effects in the whole cell recording experiment, we first examined whether the melatonin group can suppress the surge of sEPSCs caused by ischemia-reperfusion in the control group.
-Statistical confrontations depicted in fig. 4B appear incorrect, since all of them seem to be referred to the control group, while otherwise indicated in the main text (i.e. "Furthermore, in the Melatonin group, NMDA-induced currents were decreased compared to the Luzindole and melatonin group and Melatonin and CsA group", lines 150-152).
Thank you for your comment. As you point out, there is a mistake in the position of the error bars in the graph. I have corrected this and recreated the figure 4B.
-At lines 154-155 it is stated that "melatonin-induced PostC suppresses NMDAR conductance in the early stage of reperfusion and that the effect is mediated by MTs". Results indicate no change in NMDAR conductance: no single-channel analysis is here reported to demonstrate that. However, here Authors actually show a reduction in the whole cell NMDA current, that is enough to suggest an involvement of NMDAR in the melatonin-induced PostC. Reference to “conductance” should be avoided and modified accordingly.
Thank you for your comment. We avoid reference to “conductance” and corrected it to “activity”. (Line 163)
In addition, CsA abolishes the inhibitory effect of melatonin on NMDAR activity.
-Line: 352: "Opening of mPTP requires a low voltage and high Ca2+ load". This is unclear: mPTP opening occurs in depolarized mitochondria, where transmembrane potential is low and IMM voltage is close to zero (i.e higher voltage or smaller potential differential). There is confusion between decreasing voltage (i.e. to more negative values: repolarization or hyperpolarization) and depolarization (i.e. to less negative voltage values). Generally, in this manuscript there may be often found referral to a "depolarized mitochondrial membrane potential", that is formally incorrect and may generate further confusion.
Thank you for your comment. In our experiments on mitochondrial membrane potential, the green/red ratio obtained by JC-1 fluorescence indicates the depolarization of the inner mitochondrial membrane potential, which we describe as depolarization or depolarized. The decreasing voltage is described as the dissipation of ΔΦ. According to your comment, we corrected the sentences below. (Line 367-369)
The regulation of the two mode of mPTP opening in low conductance mode and high conductance mode is mainly mitochondrial [Ca2+] and inner mitochondrial membrane potential (ΔΦ) [61].
MINOR:
-Redundancy in lines 155-158
Thank you for your comment. The wording of the above Line has been simplified. (Line 163)
In addition, CsA abolishes the inhibitory effect of melatonin on NMDAR activity.
-Analysis in fig. 5C is not necessary: a smaller percentage change in the Fura-2 ratio after ischemia for melatonin-perfused slices, compared to control, was already shown in fig 5B, that represents a time course (more appropriated).
Thank you for your comment. We have removed Fig 5C.
-A general description of the method used for measuring changes in mitochondrial membrane potential (i.e. JC1 dye) should be added at the beginning of the results section 2.5.
Thank you for your comment. We have corrected a note on how to measure melatonin membrane potential in the results Section 2.5. (Line 204-208)
The green/red ratio, which represents depolarization of the mitochondrial membrane potential and obtained via JC1 fluorescence, began to increase at 5 min after the start of ischemic load and continued to increase until 3 min after the start of reperfusion (until 2 min after the start of reperfusion in the ramelteon group), then decreased in all four groups (Fig. 6A).
-Analysis in fig. 6C is not necessary. The smaller effect of Ramelteon compared to melatonin (fig. 6B), in depolarizing mitochondria should be discussed in the appropriate section.
Thank you for your comment. Fig. 6C is included to present the significant differences between the groups in a clear manner as well as our previous report (11). While ramelteon can only act on depolarizing mitochondria through MT, melatonin has a direct antioxidant effect in addition to its MT-mediated effects. It is speculated that the difference in the mechanism of action between melatonin and ramelteon could cause a slight difference in the changes in mitochondrial membrane potential. We added the description in Discussion part. (Line 360-363)
On the other hand, there was a slight difference in the effect of mitochondrial depolarization between melatonin and ramelteon in this experiment. This may be related to the non-MT-mediated effects of melatonin.
-NMDA was delivered by pressure puff. To standardize the molecules amount of a nano- picospritzer, not only puff duration should be kept in account, but also the injection pressure. Was the pressure kept constant? Please indicate the pressure value.
Thank you for your comment. We added the sentences below. (Line 462-464)
Apply low gas pressure (nitrogen, 4 – 6 psi) to the puff micropipette, and keep the gas pressure constant throughout the recording.
-English is often poor and the manuscript should be checked by a native speaker.
 Thank you for your comment. The manuscript was checked by native speaker again.
-Image quality is very poor and must be increased, even at this stage.
Thank you for your comment. All images have been upgraded from 300dpi to 1200dpi.
Overall, in my opinion, data enclosed in the present manuscript do not provide a significant grade of novelty: despite the interesting findings, this work does not provide a significant advancement in the field. The lack of a clear mechanism is one of the major cons. Moreover, the quality of the manuscript should be definitively improved to reach publication grade.
As the reviewer described, we cannot perform direct measurement of mPTP conductance changes on ischemic tolerance induced by melatonin in the present experiment. However, it was shown that opening of mPTP, calcium kinetics, NMDAR activity, and glutamate kinetics are related to melatonin-induced PostC. Since there are no reports elucidating the exact mechanism of the neuroprotective effect of melatonin, we believe that this study demonstrated important findings that may provide clues to elucidate the mechanism in detail. We hope you will take these points into consideration.

Round 2
Reviewer 3 Report
Authors made lot of corrections in the manuscript and the discussion of data is now more appropriate. I still find a lack in the mechanism, but the findings presented are solid. I have doubts about the interpretation of the mPTP role, however I have no further requests or objections.